# Dietary Energy Partition: The Central Role of Glucose

**DOI:** 10.3390/ijms21207729

**Published:** 2020-10-19

**Authors:** Xavier Remesar, Marià Alemany

**Affiliations:** 1Department of Biochemistry and Molecular Biomedicine Faculty of Biology, University Barcelona, 08028 Barcelona, Spain; xremesar@ub.edu; 2IBUB Institute of Biomedicine, University of Barcelona, 08028 Barcelona, Spain; 3CIBER Obesity and Nutrition, Institute of Health Carlos III, 08028 Barcelona, Spain

**Keywords:** diet, energy metabolism, glucose, body energy interchanges, inter-organ energy relationships, handling of dietary lipids, energy storage, dietary protein as energy substrate, disposal of excess nitrogen

## Abstract

Humans have developed effective survival mechanisms under conditions of nutrient (and energy) scarcity. Nevertheless, today, most humans face a quite different situation: excess of nutrients, especially those high in amino-nitrogen and energy (largely fat). The lack of mechanisms to prevent energy overload and the effective persistence of the mechanisms hoarding key nutrients such as amino acids has resulted in deep disorders of substrate handling. There is too often a massive untreatable accumulation of body fat in the presence of severe metabolic disorders of energy utilization and disposal, which become chronic and go much beyond the most obvious problems: diabetes, circulatory, renal and nervous disorders included loosely within the metabolic syndrome. We lack basic knowledge on diet nutrient dynamics at the tissue-cell metabolism level, and this adds to widely used medical procedures lacking sufficient scientific support, with limited or nil success. In the present longitudinal analysis of the fate of dietary nutrients, we have focused on glucose as an example of a largely unknown entity. Even most studies on hyper-energetic diets or their later consequences tend to ignore the critical role of carbohydrate (and nitrogen disposal) as (probably) the two main factors affecting the substrate partition and metabolism.

## 1. Introduction: Diet and Its Use in Energy Metabolism

At present, two main lines of study focus on metabolic acquisition, distribution and use of food energy to maintain efficiently human body functions. (1) The analysis of foods and diets, which largely uses centuries-old methodologies, and its subject of study is the relationship to diet of the body in relation to health and disease. (2) The analysis and function of cell and molecular effects of diet has developed during the last 8–9 decades, using the high pace of methodological advances and accumulated knowledge at the molecular and cellular level, with important inroads into the regulation of the cell, organ, and body homeostasis. Unfortunately, both lines run parallel, and there is not enough shared—connecting—knowledge to use the enormous amount of knowledge accumulated to help explain the direct relationships of dietary components with metabolic modulation and regulation, despite the paramount importance of this nexus. The complexity (and variability at all levels) or the problem hampers any approach to the health (and survival) problems related to diet.

In this focused review, we intend to present a few critical questions that may help to bridge the gap between these two lines. In most cases, the lack of hard research data is surprising [1]. The excessive narrowness of these study goals is even more severe [2]. The forfeiting of a large mass of knowledge accumulated just before the present century [3], and limited use of quantitative approaches [4], effectively breaks the continuum of research and knowledge that has brought us to the scattering of “modern” discoveries [5], in addition to the unjustified predominance of easily available (despite their complexity) recent techniques to obtain knowledge (and publishable material).

Because of its central role in metabolism (and physiology) and its assumed predominance on diet, we focused this lineal study on glucose, its dietary sources, its metabolic utilization, regulation and –specially– on its real “essentiality” and required supply.

## 2. Why Glucose?

All heterotrophs need to periodically ingress chemical substrates to oxidize and extract enough energy to sustain their survival against predators and the scarcity of nutrients, and to allow for the maintenance of the species and the intergenerational transmission of information. This is a basic biological rule, from which, obviously, humans and other placental mammals cannot escape. Today, the methodology to produce, prepare, and ingest a variety of foods is enormous, but we continue to retain most of the essential metabolic pathways inherited from our archaeal ancestors [6]. In the end, the main—and overwhelming—process to obtain usable chemical energy is photosynthesis, which produces D-glucose, and from that, an enormous number of organic molecules which later we use to build, protect, and provide energy to maintain our structures and functions.

Glucose is far from being a “perfect” substrate [7,8] in the same way as oxygen [9] is, but both are widely available, and coevolution allows us to continue using them just for biologic economy reasons. Both are materials dangerous for cells to handle, but we use glucose since we are geared to its massive consumption because of its ubiquity for the essential fact that glucose is produced by autotrophs as basic tool for their own metabolism, thus “providing” us with it as a main energy staple.

We often tend to forget that glucose is a powerful reducing agent. It occurs, in part, because under physiological conditions its high reactivity is less patent. Glucose in physiological media is found in essentially five different forms, four of them non-reducing, cyclical, with an internal hemiacetalic bond (isomers α- and β- of glucopyranose or glucofuranose), and the “open” reactive aldehyde form [10] (Figure 1). These molecular species are maintained in equilibrium, which favor the predominance of the α- and β-glucopyranose forms [11]. They interconvert spontaneously at a relatively slow rate, which can be accelerated by the enzyme mutarotase [12]. These changes necessarily require going through the intermediate step of open aldehyde form [13]; during this brief period of high reactivity, glucose may combine with amines (i.e., in proteins), initially forming Schiff bases, but can bind also to a number of other molecules [13]. In consequence, this bound glucose is “lost” as a substrate for energy and growth, often becoming, instead, a hindrance to the stability and function of diverse proteins [14].

Most enzymes (and transporters) acting on glucose (and on some of its essential derivatives, such as glucose-6P) require glucose to be in the β-glucopyranose form [15], favoring its priority utilization because is in this form when the proximity of the -OH in C1 is closer to the C4-C5 pair, allowing the oxidation of the chain from the anomeric carbon. This selectivity, however, produces delays in glucose availability through the interconversion of (mainly) the α-isomer, predominant in plasma [11,13], that is non-usable “directly” in a number of metabolic processes. This limits the real-time access to glucose and thus its truly physiological concentration [10]. This difficulty has not been metabolically circumvented through evolution and has been massively ignored from the methodological point of view by researchers, as proven by the common but only approximate quantitative specific analysis of glucose using enzymatic methods [11].

Glucose, however, has considerable advantages as a primary energy-substrate source: it can be easily converted in two 3C fragments even under anaerobic conditions (via the classic Embden–Meyerhoff pathway), well preserved along evolution [6]. This quality is seldom recognized, especially considering the growing awareness on the perils of oxidative free radicals generated from oxygen metabolic interventions [9,16].

## 3. The 3C Fragments, the World around Pyruvate

Splitting one 6C unit (i.e., glucose) yields a variety of (essentially) 3C molecules: pyruvate, L-lactate, glycerol, alanine, serine, and phospho-enol-pyruvate (PEP), used for energy or synthetic pathways. 3C fragments are also generated from a number of key pathways in addition to straight glycolysis: the pentose-phosphate pathway, hexosamine metabolism, the Leloir (galactose) pathway, the catabolism of fructose and other monosaccharides as substrates, the incorporation of glycerides-glycerol, lactate, and a number of amino acid hydrocarbon skeletons. Not all three-carbon metabolites can be included in this 3C list. The clearest examples are propionate (yielding succinyl-CoA, see Section 7) and D-Lactate, a significant component of bacterial fermentation (including our microbiota), found in some foods. Its degradation is difficult [17] but does not proceed directly via pyruvate. In this review, we will reserve the term “lactate” for L-lactate as in most metabolic studies.

Pyruvate is the key 3C fragment from which the core of intermediary metabolism establishes the use of diet (or reserves/turnover) substrates (Figure 2), via direct oxidation for energy, used as building materials for biosynthesis or transport between pools (cell, organ, body) to achieve energy homeostasis and efficiency in metabolism. Pyruvate is also a main source of oxaloacetate (OAA) to regulate the TCA (tricarboxylic acid) cycle, i.e., the Krebs cycle (the second one) [18], or regenerate 6C units via gluconeogenesis. These processes require a critical control of the crossing of the mitochondrial membrane [19].

Pyruvate helps maintain the redox state of cytoplasm via interconversion to lactate (lactate dehydrogenase) [20] as well as the transport through the cell membrane [21,22]. In any case, the main drain of pyruvate is the formation of acetyl-CoA via oxidative decarboxylation by pyruvate dehydrogenase [23]; this way, a 3C fragment becomes a 2C fragment and CO_2_.

The regulation of glucose fate and circulating levels has been studied exhaustively, and it has been found that the alteration of these processes may cause serious harm to energy homeostasis. Glucose is strictly controlled by a huge number of regulatory agents, including insulin, glucocorticoids, glucagon, intestinal peptides, catecholamines, cytokines, testosterone, estrogens, etc. [24]. They regulate glucose splanchnic production [25] and utilization [26,27,28]. Glucose modulates the production and secretion of insulin [29] and, indirectly, its hepatic inactivation [30]. However, insulin resistance affects very directly glucose uptake and metabolism, because of changes in insulin tissue receptors and signaling [31]. On the other side, the 3C fragments (mainly pyruvate, lactate, glycerol) are regulated within the cells, but not as extensively as is glucose in their inter-organ relationships. 3C substrates, such as lactate or glycerol can be easily taken up by a wide diversity of cells, even using less specific (or quite different) transport systems [32,33].

When the cell takes up lactate, it can immediately provide NADH in the cytosol, and then follow a number of diverse paths to provide C or energy (via conversion to 2C). In comparison, after a strongly regulated maintenance of its circulating levels, glucose is transported into the cell, converted to glucose-6P, and then it can be used for the production of 3C (and eventually 2C) through well-regulated mechanisms. The main potential inconvenient for plasma-carried 3C (specifically lactate) is its charge (a proton) and its reduced state, which necessarily has to be corrected via production of cytosolic NADH, but this is the same problem glucose generates at the level of triose-P dehydrogenase. In any case, glycerol or pyruvate do not comport these hindrances. It is well known that a high number of tissues use lactate for energy under exercise [34], limited insulin resistance [35], or as a cryptic “universal cell fuel” [36]. The brain is an active user of lactate [37] and glycerol [38,39]. Liver is a net lactate user, largely for gluconeogenesis [40,41]. White adipose tissue (WAT) can take up lactate for lipogenesis [42] or produce it from glucose in large amounts [43,44]. The intestine also provides 3C substrates, in part as a product of digestion [45]. The transfer of energy to tissues via 3C has been found to be much higher than often considered [46,47], constituting a clear alternative to intact glucose [47,48]. Under conditions of excess glucose availability, its conversion to 3C eases the pressure over the regulation of glycaemia and allows for the direct use of its energy via 3C [48,49], in a way comparable to the “pre-preparation” of fatty acids (2C_n_) fragments to plasma-soluble ketone bodies (2C_2_ fragments). The relatively lax control and ease of direct metabolic incorporation allows for a loosely regulated use of 3C anywhere. This possibility is extensive to the nervous system, which can use 3C fragments to a large extent as a source of energy in addition to glucose (or as its substitute) [50].

Figure 3 shows the relationship between dietary nutrients and the 2C and 3C substrates pools. In addition to the carbohydrate paths shown in Figure 2, a large part of amino acid hydrocarbon skeletons can be incorporated to the 6C→3C pathways (largely glycolysis and gluconeogenesis) but also as intermediates of the TCA cycle (which eventually will result in OAA and then to a 3C fragment used for energy or synthesis. Lipids, essentially triacylglycerols (TAG), also provide 3C (as glycerol), but their carbon is essentially structured in 2C units (acetate, acetyl-CoA) or—massively—in 2C_n_ chains, such as fatty acids.

## 4. Are There Glucose Specific Requirements? The Peculiarities of Erythrocytes and Nervous System Cells

Glucose (in general terms, carbohydrate) is required as a main dietary macro-component for humans. [51], also probably being an essential component of diet, at least by default [52]. The commonness, diverse forms, and high proportions of carbohydrates in many diets has masked this condition [52]. Nevertheless, the extended use of high-fat and high-protein diets resulting in inflammatory responses and metabolic alterations [53] points to the requirement of a sufficient amount of 6C substrates (preferably as 6C_n_ polysaccharides) in any normal health-sustaining diet, in proportions of 6C_n_ not different at present from our early hominid ancestors [54].

The notion that the brain only uses glucose as a substrate remains alive in many texts and studies, largely for the disproportionate basal consumption of glucose by the brain in comparison with most other organs [55]. However, the widespread existence of intercellular lactate-glucose micro-cycles between glia and neurons suggests otherwise [56,57]. The main substrate for brain energy changes with development [58], but the nervous system continues using a large proportion of blood glucose under standard conditions [55], whether this is a consequence of glia cells (i.e., astrocytes) producing lactate from glucose to feed neurons [59] or the neurons themselves using glucose directly is a question not yet settled [60,61]. The intensive utilization of lactate and glycerol by the brain as a whole has been known for more than half a century [38,62] and may be part of a system of protection against the chemical dangers posed by glucose reactivity.

From the earliest stages of development, the nervous system has specific needs for some nutrients, such as glucose, as key source of energy [63]; oligosaccharides containing galactose and other sugars [64]; 3C fragments, as indicated above; specific essential fatty acids [65]; and enough amino N and essential amino acids [66]. The needs of peripheral and enteric nervous systems are less known; in the last case, we have to include the direct relationships with intestinal microbiota [67]. In any case, it is widely acknowledged that the brain has priority over the rest of the body with respect to glucose supply [63]. The extent to which the requirements of 6C can be substituted by 3C is yet an object of discussion [60,61].

The erythrocytes (at least the non-nucleated mammal—including human—ones) are also peculiar with respect to the need for glucose. They require integer glucose, since they are not able to get energy via oxidation of 3C. Thus, these cells are considered purely glycolytic, releasing lactate from glucose [68]. Since the red blood cells need little energy and are a fundamental component of the blood (which carries glucose), their needs can be easily covered by direct uptake from plasma [69]. Because their consumption is non-oxidative (6C to 3C), the return of two 3C units per glucose does not alter the global equation of glucose consumption and do not represent a quantitatively critical factor, as can be the case with parts of our nervous systems.

## 5. 2C Substrates, the Critical Conditioning of the Lack of a Glyoxylate Shunt

The triad of TAG, fatty acids, and acetate/acetyl-CoA represents decreasing levels of complexity of most energy-related lipids; their MW is related directly to their physical conditioning and metabolic usage. A critical point is their lipophilic nature, which conditions transport by the blood and the crossing of intracellular and plasma membranes. Acetate (2C) and easily metabolizable ketone bodies (2C_2_), as well as some short-chain fatty acids, are hydrophilic enough to be transported in blood or lymph and may cross easily most membranes [70]. Fatty acids (2C_n_, medium to long-chain with less than five to more than 10 2C units) are transported bound to proteins [71], and are incorporated into the cell by specific efficient binding-proteins [72] and transporters [73]. TAG are too large and insoluble to be carried by blood plasma: they are transported within lipoproteins, structured with apolipoproteins and other lipids, such as phospholipids and other classes of lipophilic compounds as minority components. The intestine and the liver initially build up these complex lipid carriers, mainly chylomicra and VLDL. They are fairly efficient systems for transporting large amounts of TAG (i.e., heavy energy packages, mostly containing 2C_n_) to provide fuel for energy and organ functions [74] in addition to the transport to and from of critical lipophilic substrates such as cholesterol, a process reserved for the smaller lipoproteins (LDL, HDL) [75].

Within tissues, large amounts of lipid are transferred between cells by using at least two mechanisms. In the first, very large (often old) adipocytes, (or those marked for apoptosis), when broken up via apoptosis or autophagia, release a large amount of micro-drops of lipid. These vesicles are taken up via pinocytosis by macrophages, functional adipocytes [76], or other cells. A selective apoptotic (and packed groups of macrophages [77]) process has been proposed as a mechanism for tissue turnover, which is applied to dead or no longer functional adipocytes [78]. This clearing process has the advantage to constrain the diffusion of cell debris and lipid drops that can cause damage to other cells, as well as limiting the clogging of individual macrophages with the remains of apoptosis [79]. The other mechanism is the use of exosomes or vesicles, which can transfer signals, proteins, nucleic acids, but also TAG between cells [80,81]. This (incomplete) list represents a wide panoply of mechanisms for transfer of 2C-based substrates, primarily between the splanchnic bed to peripheral tissues, for storage or direct utilization.

The question of transport between cells and tissues is critical for the supply of energy on time, in the most efficient chemical form for immediate use, and with a sufficient transfer rate of to cover the needs. This is, probably the main reason why there are two main systems for sending energy substrates to the tissues via blood: a) lipophilic (2C-derived, energy-dense but large and difficult to handle) and b) hydrosoluble substrates, represented by glucose and 3C, but also the smaller 2C components soluble in plasma. The logistic advantages of having both systems rely essentially in the highly dense energy TAG as substrates sent elsewhere for heavy needs, and a variety of other metabolites, usable directly by the cells faster to supply in a continuous way, and easily subjected to immediate regulation.

The irreversible path of 3C to 2C is carried out by the pyruvate dehydrogenase complex [23] (Figure 4), which is highly regulated [82,83] and controls the entry of 3C into the mitochondrion [84]. We can use glucose to build fatty acids, but the reverse (regular fatty acids to glucose) is not possible because of this critical step at the confluence of the 2C and 3C worlds. This irreversibility is typical of animals, since plants and other phyla possess the “glyoxylate shunt” that we do not [85]. Probably, we lost this shunt along evolution because the main source of 2C to feed the TCA cycle and fatty acid synthesis remained glucose-derived pyruvate.

The 3C→2C path irreversibility has important consequences for glucose and energy partition: while 6C→3C and 2C_n_→2C relationships are reversible, 3C→2C is unidirectional. In plants, fatty acid reserves can be converted into 6C [86]. However, in humans and other animals, glucose (or other convertible 6C) is needed to provide the 3C fragments used in many metabolic paths, since despite the possible excess of 2C, no 3C derivatives can be obtained from them. This leaves us with an inefficient system to obtain energy from 6C when converted to 2C, and later oxidized via TCA cycle. An excess of 2C can be corrected only either storing it as fat or through its complete oxidation, whilst any excess of 6C or 3C can be used by a wider range of metabolic pathways, or ultimately derived into 2C.

## 6. Handling Glucose, Energy Homeostasis, Substrate Cycles, and Distribution

The reversible 6C→3C glycolytic conversion, and the easy entry of 3C in most cells results in a strategic advantage of distribution, limiting substrate buildups whilst maintaining their full homeostatic availability. This process is sustained by the existence of substrate cycles within adjoining groups of cells in a given tissue, as is the case of muscle [87] or the brain, between glia and neurons [56]. The substrate cycles between different organs are widely known, e.g., the Cori cycle [88] or the glucose-alanine cycle [89], in which exercising muscle converts glucose to 3C, later taken up by the liver, prompting gluconeogenesis (i.e., 3C→6C) with the overall result (Cori cycle) of transfer of reducing power from peripheral tissues to the liver [90]. In this case, the 3C→6C interconversion helps transfer reducing power (or 2-amino groups in the case of alanine) to the liver for their reutilization (Figure 5) or disposal. This inter-organ coordination prompts a more efficient use of energy and oxygen (and 2-amino N), helping to maintain the levels and availability of substrates.

Most of the inter-organ substrate cycles were described as mechanisms preventing the dangers of reducing power or 2-amino N accumulation in muscle or other peripheral tissues under conditions of active use of glucose or amino acids (i.e., during exercise) for energy; they always present a time-delay component for maximal effectiveness. The exportation of 3C carrying the excess reducing power or N to the liver has the additional advantage of allowing the return of these 3C to the blood as glucose, completing a cycle. However, these “cycles” usually work, in real-time, as simple vectors for transfer—“open interrelationships” (Figure 5), making full use of the differential organ oxygen supply to transfer a reduced 3C (i.e., lactate) from active muscle to other muscles or the liver to regenerate glucose, resulting in a chain-transfer of reducing power [91]. This situation can be sustained only when glucose is not in excess (in the liver), since it may block gluconeogenesis [92], but the excess reducing power can be easily corrected because of the high liver oxygen supply, removing lactate immediately or after a delay and even after a buildup of acidosis [93]. The maximal “reduction” of open cycles can be found in the lung, which scarcely uses glucose [94] but uses blood protons (as the liver do) exported by relatively hypoxic tissues, using their energy at the mitochondria and thus helping limit acidosis [95].

A special situation is that of adipose tissue; it has been found to generate large amounts of 3C from glucose [48]. This process has been suggested to help lower glycaemia (and its associated problems) under conditions of excess substrate availability [96]. The extent of this transformation is considerable, especially in the mesenteric adipose mass, which receives the mixed 3C-6C results of digestion before sending them via portal vein to the liver [49]. This peculiarity is not limited to adipocytes, since the adipose stromal cells act in the same way: a practical anaerobic glycolytic conversion of 6C to 3C even under full oxygen availability [49]. This may be considered either as another contribution to regulate glycaemia or as an alternative to produce 3C fragments (essentially lactate and glycerol) in massive amounts to supply ready-to-use energy to most organs (including the brain) to circumvent the regulatory difficulties of glucose utilization under situations of excess substrate and/or insulin resistance [36,43,47,96]. The ample use of 3C as main energy substrate agrees with this interpretation.

## 7. Enter the N: Amino Acids, the Question of Diversity of Pathways, and Our Deep Lack of Knowledge of Them

Curiously, and despite their primeval importance, amino acids seldom are considered quantitatively important substrates for humans in analyses of overall energy metabolism. The most used explanations for this ellipsis are:The often incomplete hydrolysis of dietary proteins [97];Intervention of microbiota through transformation/catabolism or even synthesis of amino acids [98];Further transformation/catabolism by the intestine and the liver before overall distribution [99];Sparsely known real catabolic pathways for many amino acids [100,101,102,103] in humans;The absence of specific amino acid/protein reserves;The lack of knowledge on the regulation of essential amino acids;The gross differences between measured calorimetric pump energy content of proteins and the real energy obtained in vivo from amino acids according to the known (or assumed) metabolic pathways [103];The assumed relatively small amount of actually oxidized amino acids derived from the diet compared with carbohydrates and lipids;The uncertainty on how the N of amino acids is processed and excreted [104], which directly affects the estimations of dietary amino acid use [105];The “diversity and complexity” of amino acid catabolism;The common absence of data on diet protein amino acid composition.

In fact, the main question for eluding amino acids from most studies is the lack of knowledge. We do not know yet (in humans) the complete catabolic pathway of several essential amino acids or the implication and compartmentation of the catabolism of amino acids yielding 1C, 2C, 3C, and 4C/5C intermediates of the TCA cycle. To complicate further the analysis, we have only an approximate knowledge on how the dietary protein (in fact, the amino acids resulting from their digestion if assimilated) translates into metabolizable energy.

The role of a few amino acids in the maintenance of glycaemia via gluconeogenesis [89,106] is well known, but the focus is now on branched-chain amino acids [107,108]. These amino acids are catabolized to 2C fragments (leucine and isoleucine) and propionate (valine and isoleucine) [109] (nor a 3C but an important anaplerotic precursor of 4C fragments [110]). Some amino acids (such as alanine or serine), when oxidized, i.e., after their amino moiety has been removed, yield directly 3C (pyruvate). Others (i.e., leucine, lysine) are mainly broken down to 2C, but most are oxidized to intermediate metabolites of the TCA cycle (e.g., glutamate, aspartate, threonine). A simplified scheme of the fate of hydrocarbon moiety of amino acids is shown in Figure 6.

The amino acid hydrocarbon skeletons can, thus, provide 2C or (in a higher overall proportion) 3C, but also the high-energy containing 4C and 5C fragments, which, eventually will be converted into 3C along the TCA cycle, hence their importance for gluconeogenesis under conditions of severe energy stress. However, the use of 2C for 2-ketoacid synthesis is almost nil, widening the 2C-3C chasm. In fact, the variation of amino acid catabolic pathways results in a simplified handling of their complexity—losing in some cases energetic efficiency in the process—for gross energy purposes (as shown in Figure 6) than usually believed. As to the aspects of regulation cited above, we are yet in a serious state of ignorance, with few exceptions.

## 8. The Dangerous Duality of Needed 2-Amino-Nitrogen Preservation and Its Ill-Understood Disposal Pathways

From an evolutionary point of view, our design evolved to preserve 2-amino N, since we cannot significantly use other sources of 2-amino N than amino acids themselves (obtained from dietary protein). All amino acids are synthesized by plants (and other phyla) and used to build their own proteins. We use (eat and digest) these same proteins to obtain the amino acids we need to form our self-proteins and other N-containing molecules (e.g., purines, pyrimidines, porphyrins, etc.). All our amino acids have been obtained (from the diet) either preformed—e.g., essential amino acids—or rebuilt by us from available hydrocarbon structures and 2-amino N suppliers, i.e., amino acids from plant or animal (formerly “plant”) sources. Since we are omnivores, the dietary protein is just another source of energy, used, for this purpose, in proportions that depend on their availability [111,112]. Thus, the complex safety measures established to retain amino acid N necessarily should interfere with their utilization for energy. Nevertheless, it is obvious that essential amino acids are continuously oxidized for energy irrespective of their source: if there are not changes in body protein (in addition to the “obligatory” protein losses: Box 1), the amount of these amino acids ingested in the diet should tally their oxidation, since they cannot be stored.

Box 1Forms of excretion/loss of nitrogen in humans.
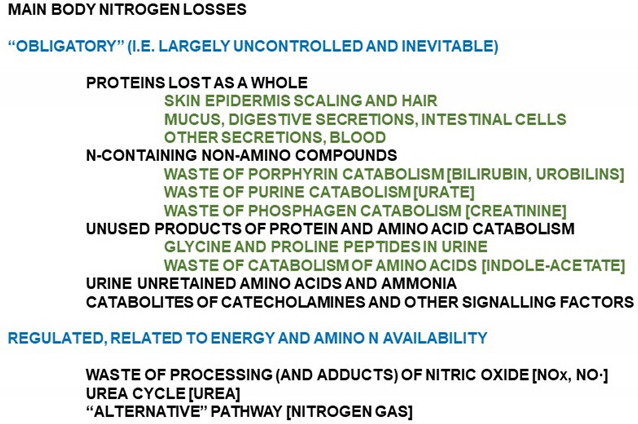


The main, basic studies on our handling of diet amino acids, protein synthesis, and recycling of amino N were done essentially more than half century ago, and they were invariably focused on the ways to preserve N, the mechanisms to survive under conditions of starvation [113], dietary protein deprivation [114] and, globally, malnutrition [115,116]. Amino acids as energy staple have been, since, considered “secondary” (or at most “complementary”), and their assumedly “complex” metabolism was oversimplified around their possible use as gluconeogenic substrate under glucose deficits or generators of ketones (ketogenic) during starvation [117]. The epidemic nature of obesity, developed thereafter, and the need to find ways to cope with its ravages were centered on glucose and fats, a situation that continues with limited global interest on amino acids as energy substrates under conditions of plenty [118,119]. The key problems that remain and prevent further advances in this direction are the diversity of metabolic pathways used to oxidize the hydrocarbon skeletons of amino acids, the relatively unknown mechanisms to retain essential amino acids, and, especially, the not yet clarified variable fate of the N moiety of amino acids during their catabolism.

The canonic mechanism of excretion of the N moiety waste is the urea (or Krebs–Henseleit) cycle [120] (the “first” Krebs’ cycle [121]), a very peculiar pathway between cytosol and mitochondria. It may also harbor a shunt to generate nitric oxide (NO·) and citrulline from arginine [122]. The daily losses of N via the NO· pathway are a fraction of the overall N excretion, essentially made up from nitrogen oxides released through respiration [123] (a few µg/day) or further oxidized to nitrite, nitrate, and other nitroxylated compounds [124,125], which are excreted via saliva, urine, and stool [126,127]. However, there is proof of an additional, quantitatively significant, excretion of N as gas through respiration [128,129] that is supplementary to the standard and well-controlled urea excretion. When analyzing in detail N balances in experimental animals, the amount of N ingested is higher than the sum of the N accrued in the body and the N excreted and accounted for [130,131] (Box 1). The proportion of this “nitrogen gap” (assumedly N_2_) is in the range of 5–15% of all nitrogen ingested and not accrued (i.e., it is actually excreted), and its amount is related to diet and energy status [128,130]. It has been found that this N_2_ loss is related to arginine metabolism [131] (Figure 7).

In fresh-water plants [132] or fish [133,134] living under high (cytotoxic) ammonia concentrations, the levels of NO· are high, and the production of NO· is increased several fold on exposure to ammonia, assumedly potentiating its oxidation [134]. The most suggestive process is that of the anammox (anaerobic ammonia oxidation) bacteria [135], which assumedly use NOx, probably via NO· to oxidize NH_4_^+^ coupled to the reduction of NO· [136].

Both ammonia and nitrite or nitric oxide can react spontaneously in an exothermic reaction, yielding N_2_ and water [137], provided that ammonia is protonated to ammonium and the extra electrons are removed
4 NO + 2 NH_4_^+^ → 4 H_2_O + 3 N_2_ + 2 e^-^(1)

This is a reaction very similar to the integral oxidation of ammonia with nitrite [138],
NO_2_^-^ + NH_4_^+^ → N_2_ + 2 H_2_O(2)
which can be reproduced very easily in the laboratory with minimal means. N_2_ is also produced by the reaction of nitrite with amino acids at low pH [139]. The abundance of nitrite produced in the inactivation of NO· could justify a small part of the production of N_2_, in part via its reaction with amino acids [140]. However, this process has been studied under conditions seldom found in a living human [139]. Nitrite may react more easily with ammonia, but their low circulating levels [141], and the small area of diffusion of newly-created NO·, in the µm range, may prevent this reaction taking place, except, perhaps, in the gut lumen, where the levels of ammonium and NO· (but not the environmental conditions) may be higher [142,143]. However, the fact that in axenic rodents the excretion of N_2_ is unchanged [128] hint at the existence of a less “occasional” and regulated source for the production of N_2_.

The mechanism of nitrogen gas synthesis from ammonia used by anammox bacteria suggests a possible pathway in which this oxidation is carried out in two steps [137]:NO·+ NH_4_ + 2 H^+^ + 3 e^-^ → N_2_H_4_ + H_2_O(3)
N_2_H_4_ → N_2_ + 4 H^+^ + 4 e^-^(4)

Evidently, the hypothesis we presented in Figure 8 with respect to the postulated origin of N_2_ from the mitochondrial reaction of ammonia and NO· has a serious weak point, since, as far as we know, the enzymes that allow the oxidation of ammonia by NO· (under anaerobiosis) are exclusive of anammox bacteria. However, the possibility of hydrazine being an intermediate step in the process could not be fully discarded given its limited toxicity (it has been used for the treatment of tuberculosis and cancer [144]) and its widespread effects on amino acid metabolism [145].

Ammonia and nitric oxide are both produced in significant amounts in the mitochondria: ammonia through—mainly—glutamate dehydrogenase [146] and glutaminase [147], and NO· via NO· synthase [148] (Figure 8). One can speculate that within the mitochondria, the reaction 1 can take place through a regulated mechanism. The main problems for this hypothesis to be confirmed are the so far unidentified catalyzer system, and the question of the low solubility of N_2_ in biological fluids but not on fat tissue, several-fold higher than in plasma [149,150], to carry the gas away to be exhaled in respiration. This type of shunt is, probably, the base for the known excretion of N_2_ gas under certain circumstances (high arginine, high protein) resulting also in excess ammonium in the mitochondria [151]. This uncharted metabolic path is affected by diet, sex, and excess energy and amino-N availability [128,130,131], and constitutes an example of our extremely low level of knowledge on the mechanisms and factors that govern substrate energy partition. This uncertainty affects seriously any quantitative study on this subject.

## 9. Reserves, a Highly Reserved View

We need (and thus maintain) reserves of all possibly scarce materials we need: essential metals, ions, sulfur, selenium, some vitamins, and especially 2-amino-N. Evidently, we also store energy. The balance between reserves, their type and the problem of carrying them as part of our body weight has been extensively studied, but often, we tend to maintain excessive long-term reserves of lipid (2C_n_). We retain a lower amount of 6C_n_ reserves (glycogen) for hypoglycemic emergences, and we also keep a small supply of phosphagens for immediate support to the first line of response (ATP).

Metabolic syndrome (MS) and similar conditions related to excess of energy supply create havoc with the management of reserves, since the unmanageable excess of 2C tends to end as lipid, which is massively stored for lack of alternatives. This problem derives largely from our taste for fat [152] (inherited from our ancestors because of its high energy density), perhaps helped by our ingrained memories of lactation [153] and the availability of “tasty” lipid-rich foods [154]; but it is also derived from an unbalanced supply of energy (especially in excess) of 2C vs. 3C as a main energy staple [155].

Our main (from a quantitative energy content point of view) physiological reserves are made of fat because WAT contains up to 80% TAG (in the range of 30 MJ/kg) [156]; thus, a few kg of WAT may provide enough energy for our basic sustenance for months (Table 1). Its main advantage is the high energy densities of fat and the adipose tissue holding it, especially when compared with polysaccharides. Our reserves of glycogen are small [157] and were devised to sustain glycaemia for a short time, since glycogen contains about half the energy than fat (on a dry weight comparison basis). In addition, it is highly hydrophilic and retains a large amount of water (added weight, limiting its accumulation in significant amounts) [158]. A high weight (including water) with respect to energy content limits movement excessively in exchange for short-lived reserves; thus, the main reserves, the more effective in terms of energy/weight are lipids. However, their real effectiveness depends on the body mass of protein [159].

The main theoretical problems posed by body fat reserves are essentially two: a) its use as storage of energy may derive into being a 2C dump when energy intake is excessive, driving to obesity, inflammation and MS [164]; and b) we need, specifically, glucose/3C for inter-organ supply of energy. Thus, sources of 3C should be taped to preserve our homeostasis under conditions of severe scarcity; even when there is a large excess of 2C reserves available, we keep needing a steady supply of 3C, which often requires the sacrifice of protein [165]. In practical terms, the main 3C “reserve” of the body are the proteins (irrespective of their functionality). However, since we do not have protein reserves as such, we have had to rely on misadjusting the overall turnover rates to degrade faster and synthesize less protein [166], draining the amino acids freed to generate 3C, especially during starvation [167].

The data shown in Table 1 are only general approximations, but the figures of storage substrates and circulating compounds used with different intensity as nutrients throughout the whole body illustrate clearly the enormous difference between the make-up of 2C substrates and the 3C ones that may be used for functions other than the generation of energy through the TCA cycle. The 2C/3C ratio for all added-up energy reserves in normal weight healthy adults is about 24-fold higher when considering body reserves than when we analyze the nutrients carried by blood plasma. This highlights the differences (even with all logical caveats considered) between what is the homeostatic internal medium composition, akin to our immediate substrate supply needs, and what we have retained. The weight of 2C is overwhelming.

In Table 2, we have intended to obtain a value of the C2/C3 ratio in the human diet. The problem is the absence of data (especially on protein composition of the enormous varieties of diets) We analyzed the C2/C3 ratio for a “theoretical” adult healthy human eating a standard diet, such as those being recommended for a “healthy nutrition” by most specialists. The results were again shocking: the final ratio of the standard diet was in the range of 1/5th of that of our “normal” body composition, and it was more than five times higher than the plasma nutrient ratio. These differences stress an innate deficit of 3C yielding substrates, necessary to maintain energy homeostasis with respect to the energy stores, strongly decanted toward the 2C supply, making a large part of these reserves not physiologically usable and, in fact, only “dead weight”. This unchecked domination of 2C sources hint that our standard diets are probably inadequate, chosen more because of factors different from the best metabolic adequacy (taste, availability, social or historical factors, etc.). The recommendation of such types of diet adds to the problem they pose, since recommendations are based on population statistics with no ‘controls’ to refer to. They are not based on sound biochemical metabolic comparative data, simply because they have not been (sufficiently) investigated. The consequences are crystal-clear: a growing pandemic of diet-borne disorders, essentially undiagnosed, not treatable and affecting a large (and growing) proportion of our Society.

It is important to add an additional commentary on the methodology we use at present with diet (food) analyses. Despite the considerable difficulties, assumptions and techniques available, the classical study of Atwater [172] remains the “current” basis for our energy intake/expenditure calculations. In 12 decades, science has changed considerably, and precision is one of the critical points for accepting such widely used data. A recent critical analysis of the methodology used by Atwater but applying present-day criteria [173] showed that a significant deviation of the original data exists, and that we urgently need methodology more adapted to today’s conditions.

The dynamic situation of lipid reserves is clear and understandable. What is not is the reason why some body fat seems impossible to mobilize even using very-low energy diets [174,175] even with metabolic situations close to starvation, which resulted in scant (if any) mobilization of part of the fat reserves [176], often in spite of real body weight losses (water, protein, minerals). The extra weight and the inflammation accompanying the excess of adipose tissue fat are severely detrimental for the health and normal energy handling, and are maintained without any apparent reason [177].

Our energy metabolism overall can sustain, in a large proportion, the substitution (as energy substrates) of part of carbohydrate (3C-givers) by lipid (2C-providers). Carbohydrate may be compensated in part by protein [178]. However, the N moiety of protein and some amino acids could not be substituted (but we can lower our needs by modulating turnover rates) [179]. These compensations include the use of internal reserves (mainly lipid) but also, in part, the lowering of the body protein pool [165]. These measures do follow well-established priorities in the use of energy substrates (or, in the end, any substrate) ingested, and, when insufficient, compensated with the use of reserves [180]. The order of precedence in consumption (that of preservation is just the reverse) [180] is essentially:1 Lipid (2C)→2 Carbohydrate (3C)→3 Protein (amino N)

Diet takes precedence in their use for energy over reserves, but changes in the diet may modify some of these evolutionary rules of precedence, thus protein is needed for lipid oxidation and to prevent its storage [181]. These mechanisms were developed, and refined with efficient regulatory mechanisms, to enhance survival by maintaining energy homeostasis under conditions of insufficient food availability (including starvation), and incorporating the hypothesis of thrifty genes and their possible incidence in the development of diabetes/obesity [182,183]. However, we have not established a metabolic process (spendthrift) to cope with a simultaneous excess of energy and of all main nutrient classes in the diet. No process has evolved to solve this situation because this is a peculiarity, affecting practically only our species, which appeared in a very short time (in evolution terms) [183]. The trend is fueled in part by the setting of epigenetic traits favoring the development of obesity, despite its real deleterious effects [184,185]. We have no historical evolutionary memory on how the problem of excess can be handled, since our tools are useful only to counter scarcity, and thus, the mechanisms/processes applied are often not only inadequate, but self-defeating and harmful.

## 10. The Triad of Main Energy Sources and How Sorting of Substrates Seems to Proceed

We ingest food for energy and growth/maintenance. Evidently, a large part of the main substrates (amino acids, fatty acids, sugars and intermediate metabolites, as well as most essential components) are used in connection with turnover strategies of our own living matter, in order to maintain all mechanisms in their prime efficiency. However, from a theoretical (quantitative) point of view, intake should match excretion and energy loss (assuming that there is no real change in our mass) [186,187]. Thus, the main types of substrate we ingest for energy (with all provisos stated above) belong to three groups: amino acids (N providers), 3C and 2C providers. After a large portion of amino acids is sieved and selected, they revert largely to “excess N” plus 2C, 3C and 4C (TCA cycle intermediates) (Figure 2, Figure 3 and Figure 6). The final proportion of 2C and 3C available during a certain stretch of time depends on digestion, rhythms etc., but we can just center the question in basic one-day periods. Under no-growth conditions (i.e., no additional storage of energy: only turnover) almost all 2C will end as a source of energy via TCA cycle. 3C have a higher variability of uses, but at last, all excess 3C is converted to 2C, which is then oxidized for energy (or stored as fat). If there is a large excess of amino acids, the elimination of N stresses the 2C/3C setup, both because of anaplerotic enhancement of TCA cycle (via 4C) and an intense pressure for growth, which increases protein synthesis and “storage” as body protein. Most of these effects/mechanisms remain rather unknown, because almost never come alone, and are largely mediated via gene expression and hormonal modulation affecting select different groups of cells.

A large part of nutrients is used with little change (if any) for turnover (e.g., amino acids, fatty acids, glucose, etc. for, respectively, proteins, TAG or glycogen), or to build-up reserves (i.e., fatty acids) or (almost all compounds) for growth and reposition of live matter lost (i.e., hair, epithelial cells or secretions). Amino acids could not be substituted from the point of view of reposition and turnover, as well as for the synthesis of other N-containing body components. Carbohydrates may be substituted, largely, by protein, but the loss of energy in the complex catabolism of some amino acids, plus the need to dispose-of 2-amino N markedly limits the extent of this substitution. On the other side, lipids (e.g., TAG, but not including the essential fatty acids) can be entirely substituted because their sole specific metabolic contribution is to provide large amounts of 2C, which we are prepared to compensate with protein and, essentially carbohydrate.

An excess of 2C, on the other side, limits the utilization of dietary fatty acids, which are then incorporated to the stored fat reserves, without previous oxidation to 2C [188]. This is an efficient way to store dietary energy, quite different from the energetically expensive lipogenesis from acetyl-CoA [189]. This 2C molecule is also in excess (limited by the availability of coenzyme A because of insufficient capacity of the TCA cycle to oxidize it, since reducing power/ATP are produced on demand. This is compounded by limited lipogenesis, restrained because of the excess of dietary fatty acids [190]. The large acetyl-CoA availability, due to its limited need, inhibits the oxidative decarboxylation of pyruvate, the critical 3C→2C unidirectional process [191], thus secondarily provoking an unwanted accumulation of 3C. In parallel, glycolysis is also limited because of insufficient 3C removal [192], which results in lower glycolytic processing of 6C, which adds to the excess carbohydrate provided by the diet together with the unneeded excesses of dietary amino-N and fatty acids (Figure 9).

The serious difficulty lies on the iterative nature of the process along time. A few days of excesses can be modulated in part by decreasing appetite and voluntary intake [193], increasing protein turnover [194], thermogenesis [195] and topping the reserves of glycogen and fat [196]. Nevertheless, these ‘solutions’ work only to a certain degree, not in a permanent way, and create a regulatory havoc that cannot be sustained indefinitely. The emergency measures only patch, do not solve the problem, it is just metabolic procrastination, and in short time deep problems begin to develop [197]; and shortly afterward, they also become chronic [198,199]. The excesses of substrates could not be disposed of, and the regular mechanisms of partition cannot function in a fluid way. Obviously, oxidation of 2C is enhanced, but what to do with the unused ATP? In a global setting, thermogenesis helps, but not at the level of each cell (largely those in the splanchnic bed). Then, the priorities list ingrained in our metabolic control systems retake their place of reference.

Save as many amino N as possible and take special actions to prevent the toxicity of NH_3_, perhaps via the alternative “NH_3_ + NO·” production of N_2_.Limit the processing of fatty acids to 2C fragments and thus dump them as TAG (with a token 3C glycerol) [200]. This process induces obesity [201], steatosis [202], and drives muscle to be essentially a fat-infiltrated 2C energy user, with glucose intolerance [203,204].The toxicity of glucose (the same occurs with the excess of ammonia) represents an immediate danger that cannot be “stored” (as are the fatty acids in TAG).

The insulin basic controlling mechanisms of glycaemia no longer function properly [205], in fact they interfere, limiting the peripheral utilization of glucose as substrate [206,207]. Hyperglycemia is a problem [208] which is in part corrected by the conjunctive tissue (including adipose tissue), which actively takes up glucose and returns 3C to the blood [96]. Most tissues may work (including the brain) perfectly using 3C (largely lactate and glycerol) [36,37,38,91]. Thus, the blockage of liver metabolism, in its ↑2C-↑3C no-win situation, is in part corrected by an adipose tissue (also inflamed, engrossed by the excess lipid storage [209]) which generates large amounts of 3C at the expense of glucose [96]. It also produces urea cycle intermediates, probably to compensate in part for the decreased capacity of the liver [210,211].

## 11. Normalization and Regulation under Excess (if Any)

Return to normalcy from the excess-driven picture of energy metabolism is difficult. The search for drugs is not a viable solution, because the problems observed have a known origin, and the severity of the damages is largely a consequence of the mechanisms established to ensure survival under scarcity. We cannot fight these mechanisms because a) we do not know them enough, b) we need them to sustain our “normal” homeostasis, and c) they have to be operative to achieve that normalcy. Right now, the main systems in use to fight obesity are—essentially, and despite their limited effect—diet and exercise; but, what type of diet? Hypocaloric?

Under starvation, the stored fat is progressively shed to last as much as feasible; our bodies adapt to lower energy, lower carbohydrate and limited protein intake diets, including the absence of fats, following the blueprints for starvation [212]. The elimination of dietary carbohydrates [213,214] and lowering of energy intake share some characteristics, because of our adaptability to starvation, but there is a considerable discussion on the proposed benefits of ketogenic diets, which, in any case, could not be generalized from epilepsy to obesity [215,216,217]. Nevertheless, even after prolonged starvation or removal of dietary carbohydrates, a significant portion of body fat remains, even after prolonged exposure. Unfortunately, in the practice and for most obese people, dietary treatments remain not effective [218].

Exercise is an alternative to increase energy expenditure with (or even without) restricted intake [219,220]. This approach is not always feasible, since obesity is syndromic with cardiovascular disease and problems of mobility [221] and, despite numerous claims on the contrary, exercise it not sufficient to revert the damages of an established MS [222], and may not improve the health status [223]. However, well dosed exercise may provide benefits to mild cases and young patients [224]. In addition, the effectiveness of exercise is also limited from the point of view of energy: limited duration, limited proportion of increase in overall energy consumed, consequences in the availability and transport of oxygen and products of substrate oxidation. In fact, exercise is better suited to maintain functionality [225] (when applicable) than to fight obesity. The chronicity of the “excess situation” described above, usually continue even when the dietary excesses cease. Too often, very low-energy dieting and exercise combined (even using additional anti-obesity approaches) are unable to eliminate the excess fat and reverse the damages already induced by (or accompanying) this excess deposition. The size of fat depots is a critical point for the severity of the disorders and a barrier to a progressive and effective removal, often leaving only the alternative of surgical modification of nutrient intake [226]. However, none of these procedures can revert the metabolic homeostasis to the situation prior to depot engrossment. Nevertheless, drastic treatments may extend the patient life span, improve the lifestyle and ameliorate a few components of the MS [227,228], but not cure it or its associated pathologies. The corollary to this exposition is our extreme resilience to change in the energy (and nutrient) equilibrium, which allows us to cope with the situation for a time and then alter, in a probably irreversible (and unknown) way, the common mechanisms for nutrient partition in a situation far from normal, which could not be sustained indefinitely. However, these changes allow a number of affected individuals to keep living despite the severe damages to metabolic processes and their control. A deeper layer of thought (and thorough research) are needed to understand how people can survive decades under MS conditions.

Probably, excess body fat is directly related to the severity of the disorder it causes: it is commonly accepted that the body mass index vs. mortality curves are U shaped [229]. However, this index is not a real indexed measurement of the body fat content, and parallel studies analyzing body fat vs. mortality showed a lineal direct relationship [230]. Nevertheless, a certain degree of obesity helps protect the patient from some cardiovascular diseases [231]. The discussion about the causes of the obesity paradox or even its existence continue, but part of the problem may lie on the inadequate use of body mass index to “measure” obesity [232], and the question of whether obesity is by itself a disease (or simply a part of a more severe disorder such as MS). The repeated references to “healthy obesity” (i.e., large body fat stores without the associated metabolic disorders of “proper” obesity) [233], have been relatively circumscribed by introducing factors such as age and gender [234]. Our group has postulated that a certain degree of obesity may help limit the ravages of type 2 diabetes by removing excess glucose and releasing 3C, being more a (palliative) consequence than a cause for this disease [49,96]. In any case, the question is not settled, and perhaps there is also a margin for optimal (and not rigidly low, invariable and universal) mass of body fat reserves other than the one used at present.

Under conditions of excess energy intake, the metabolic handling of substrates is parallel to modifications in the hormonal mechanisms that regulate energy metabolism [235], since the main system of regulation of glycaemia (insulin) has been severely damaged [236]. The other mechanisms complementary to insulin have a wider array of functions: the glucocorticoids, favor liver glucose output under conditions of stress or metabolic distress [237]. Glucocorticoids also affect the fate of the main gluconeogenic precursors, amino acids, and altering the excretion of N [238,239]. Testosterone pairs with insulin as a main anabolic hormone [240], but glucocorticoids tend to limit testosterone production and availability [241], also affecting the availability of estrogen [242], which plays a critical function favoring the oxidation of 2C (i.e., saving 3C) in females [243] via direct intervention in mitochondrial function [244,245], and also preventing liver steatosis [246]. An estrogen-derivative has been found to down-modulate the adjustment of the ponderostat, i.e., the oxidation/mobilization of lipids from adipose tissue [247,248], by decreasing food intake and maintaining thermogenesis [249]. Unfortunately, there is insufficient mechanistic information on the effect of estrogen derivatives because only recently these hormones are considered important metabolic regulators [250,251] and act not only in the sex-related way indicated by its etymology.

Substitutive testosterone treatment normalizes the glycaemia of mature and old men, a situation that is slowly being accepted by endocrinologists (diabetologists, in fact) [252,253] after being a mainstay of the treatment of MS aging-related disorders by gerontologists [254]. Its soothing effect on the MS ravages [255] is complemented by the already known effects of estrogen on energy partition [256,257], limiting the storage of lipids. Most of present-day studies on obesity (and of MS, as an afterthought) are focused on cytokine and regulatory circulating RNA types [258] and the paths they modulate, which add considerable knowledge on the possible regulation of their mechanisms of action, but not on the causes/mechanisms of the disorders or the ways to prevent and treat them. Figure 10 shows a simplified scheme of the main hormone-controlled mechanisms regulating glycaemia.

In the liver, glucocorticoids increase glucose output [237] and favor lipogenesis [259] and TAG deposition in most tissues [260]; testosterone induces the accrual of protein [261,262] and stabilizes the maintenance of glycaemia [252]. Estrogens favor 2C oxidation [263], increase oxidative metabolism in mitochondria [245], and limit lipogenesis and TAG deposition [264]. The role of these steroid hormones on the direct modulation of glucose is less clear, despite the large number of agents and effects uncovered. We already know a part of the puzzle, but there are not yet enough dots to draw a sufficiently clear line to understand their real function and help us fight the ravages of our own (effective) systems of protection of energy and protein.

## 12. Diet, Its Composition, and the Central Role of Glucose

Man’s main energy staple has been, in the last millions of years, of plant origin: seeds, starchy roots or other plant energy reserves, fruits, leafy foods, complemented with some products of animal origin. Humans have evolved in parallel to their nutrition [265]. Foods from animals have been increasing their share with time [266], and our diets are clearly adapted to our pre-human (but pervading) basic food taste types. Less exercise, better medical attention, longer lives and control of the sources of food have modified our present-day diets. Culture, unproven ideas about food have been combined with religious or social taboos, increased availability of different presentations of food and the culture-driven extreme cult to taste. We are leaving starches to consume fats [266], and animal muscle protein and collagen instead of aleurone-like plant reserves [267]. However, our digestive system and metabolic energy handling systems were adapted to a quite different diet [268], and have not been able to maintain the pace of change in eating habits occurring in the last centuries [266].Thus, a gap between digestive system/nutrient utilization and diet composition exists and keeps growing. Glucose continues being the main energy substrate, in spite of everything else, but its food form, ingestion, digestion, handling, and oxidation necessarily change with the composition of the diet. However, that of Neanderthals showed little differences with our present day optimal diet types [269]. The periodicity, the ratio of energy consumed (or needed) for a straightforward metabolic equilibrium versus that actually ingested is a powerful destabilizing factor for an effective maintenance of metabolic equilibrium. Activity, feeding, light and internal rhythms control our hormones and metabolism, helping us to adapt to the variable conditions of the environment [270]. Probably this imbalance is a main cause of the MS together to the fattening of the diet [271].

Considerable observation and experimentation have shown that amino-N should be a part of our diet (in the range of 10–15% of ingested energy) [272]; and, also, that protein should be ever present in our meals [273]. This implies that we also need to ingest an adequate supply of a few special amino acids we cannot synthesize [274]. In parallel, we also know that, today, most humans are taking in too much fat in their diets [275], often over 30% of all energy ingested [168,169]. An effect compounded by the ingestion of energy above our physiological needs [276]. There is little quantitative analysis of necessities, amounts and proportions, as explained in detail above. These approximate proportions are quite different from those of all present-day apes, even more than most of the other primates [277]. This comparison should include our earlier human ancestors, usually short in protein, and even shorter in lipids. Hints on the postulated hunter-gatherer ancestors feeding standards and behavior, and the study of human dietary variability at present suggest that we are a remarkably adaptable species [278], but in all cases, 3C takes priority over 2C, often with the help of the highest protein intake of all primates [161,162]. Evidently, within the enormous variety of human groups and their adaptation to all types of diet (the food available) and environmental conditions, it is expectable that a wide variety will be found –also– in the differential use of the main nutrients; provoked, in general, through epigenetic-driven adaptation to the foods available [279]. But most humans not living under extreme conditions maintain diets which, despite the ample variability in foods and their proportions, provide enough amino-N, 3C and 2C for healthy survival [267] (Figure 9).

The problems arise when the carefully established compensatory mechanisms for diets markedly deviate from those of our ancestors, which result inoperative or deficient by excess of energy and, especially, of some nutrients [199,205]. It seems that the Paleolithic (including ancient historical times) physiological blueprint has evolved little in regulation, despite our admirable adaptation, to occupy almost any ecological niche conceivable on Earth, and thus, diet-provoked “inflammatory” diseases appear and remain unchecked [280].

Variable food supply, including in some cases its excesses and early death (for almost any cause imaginable including diseases) shortened the lifespan of our predecessors already in historical times [281]. Notwithstanding, in the present, our main health dangers seem to be related to our “excessive” success in providing food and health care to a large proportion of humans, coupled with an insufficiently fast evolutionary pace to let our bodies to adapt to this, so far, unique success in the extension of a species mean lifespan and its consequences [282]. Enter the problems of aging: until recently, a problem affecting a very small part of developed societies, that became a prime cause of health, lifestyle and social problems because of the success in expanding our longevity [282,283].

Despite all that, we have not established (in times of plenty) an estimation of the magnitude of 3C-precursor nutrients we need to consume. Essentially the question is: do we need a minimum intake of carbohydrate? [284]. Glucose is the main energy staple of our diet, and we base on 3C substrates most of our metabolic function and regulation [36,47,48]. Instead, we centered our attention on lipid (2C sources) [285,286,287], sugars as such [288], some minerals (e.g., salt), and, to a limited extent, on protein [289]. It is unclear why our flag substrate, glucose, has not been given due importance except as a marker of disease. However, the pervading idea of almost any food-derived nutrient contributing effectively to our energy sustenance, with (almost) full substitution possibilities (as energy fuels) between carbohydrates, fats, protein (and even alcohol) is deeply rooted. This belief would render superfluous to analyze if there were a dietary minimum supply of 6C-3C substrates, in the way we know that this minimum exists for amino-N and “loose” maximums are generally accepted for lipids and alcohol (a toxic substance and “pure” 2C). From the data available, the quest for lowering glycaemia and reducing body weight [290] favor the use of low carbohydrate ketogenic diets [291]. However, high-carbohydrate diets improve glucose metabolism [292] in healthy individuals, and restriction of diet carbohydrates increases the risk of cardiovascular disorders, depending on the diet energy and composition, especially in patients with disorders of glucose metabolism [293] (but not in healthy individuals [294]). Not even glycaemia is better regulated in diabetics with excessive dietary carbohydrate restriction [295]. The disarrangement provoked by the relative insufficiency of carbohydrate in some “low-carbohydrate” diets depends, essentially on four factors:(a)The metabolic “basal state” of the patients, since the results depend on gender, age and the incidence of MS-related disorders.(b)The overall energy intake with respect to the needs (or slimming goals) of the patient.(c)A diet low in carbohydrates must get the energy from either protein or lipid, the proportion and structure of these nutrients in the diet affects the incidence of lowering carbohydrate intake.(d)The type of carbohydrate present in the diet and its food associations, essentially its relationship to fiber and the mean molecular weight of the polysaccharides (including their digestibility and effects on the microbiota).

The enormous variation of results, consequences, and effectiveness of the proposals coupled to the soundness of the results from the data published, results in a near-impossibility to draw conclusions. In addition, almost all diets studied have been applied for short periods of time to small numbers of individuals with scant information even for the point a) and were done in ‘comparison’ with other different types of diets. Even the reviews and meta-analyses could not draw clear consequences from the huge amount of literature accrued on this question. There are too many factors to allow us to draw safe conclusions from partial, incomplete and not superimposable data. To face this type of analysis, a much-needed simplification is required, and reserve on the conclusions (and on their application to people) must be observed.

Diet studies largely analyze the consumption of food items (a huge, varied and socially peculiar group of products), which in addition have been cooked and mixed in varied proportions. This is “reduced” to the nutrients they contain, but this may forsake aspects such as synergisms, food structure (i.e., fiber), and the nature of the nutrient itself (fructose is as carbohydrate as amylose or a resistant-starch, for instance, and collagen-rich squid is as protein as gluten). We know what is eaten but not what is assimilated (the variable microbiota plays a critical role on this process). We need more information on—at least—what is ingested along longer periods of time.

In any case, a multilateral approximation to the data known and the line of thought presented here allows us to assume that a minimum (both in proportion and in absolute terms) of daily dietary carbohydrate does exist [284], despite the possibility to substitute a large portion of the 6C intake by other sources of 3C.

## 13. Perspectives, Conclusions, and an Urgent Call for More Quantitative Basic Nutritional Studies

There is abundant literature but a dearth of information on diets. Unfortunately, when dealing with the concept of diet industry, policy, fashion, absurdity and taste interact to produce dispersion, unproven assertions, expanded private interests and outright lies, with—unfortunately—scant material usable to advance scientific knowledge and improve the preservation of health. In most of the studies available, an enormous variability in all the factors implied can be observed, and the formal and bona fides results rarely go further than “hinting to,” with excessive room for interpretation and discordance (or even actual overall relevance). The most recent and popular case in point is the Mediterranean diet, which, in fact, has not been yet defined in nutritional terms [296] but has generated, nevertheless, a considerable number of studies on its health benefits [297,298].

Why do relatively small modifications in diet may induce so deep changes or no changes at all in functional parameters? We do not yet know enough to obtain a defendable and plausible answer. On one side, the nutritionists and dieticians usually analyze diets as a whole, use tables and measure our anthropometric (and metabolic) data and detailed food intake to calculate the needs of nutrients (translated to foods) to cover the energy expenditure, despite the fact that both factors are closely linked. On the other side, a growing number of molecular biologists analyze the specific mechanisms of control of endocrine and paracrine factors (largely those easily measurable), usually using isolated cells or cultures, since this approach is not viable in most in vivo models. Both worlds of study are eons apart, and keep publishing large amounts of deep, sometimes excellent studies on “new” signaling factors or mechanisms, and the beneficial properties of some foods. Our scientific careers have passed through different consecutive fashion periods: enzymes, hormone receptors, cyclic nucleotides, gene analysis, cytokines, gene expression, microRNAs, gene modification, exosomes, stem cells, the transcendence of microbiome, “virome/pandemic” etc. However, surprisingly nobody has devoted sufficient time and funds to study such elementary “unknowns” as how “essential” are some amino acids (and which are the paths of their complete catabolism in humans), how the excess N is eliminated (signals, pathways, mechanisms, sites), or how much 6C (largely glycosyl monomers) we need to ingest daily? What is our dependence of 3C as energy substrates? How is regulated the utilization of 2C/3C for energy (not only in a type of cells, but overall)? Where does this take place? Why there are not enough quantitative studies on how much lactate or glycerol uses the brain, muscle or heart? Is WAT a real storage depot or a secondary supplier of 3C from excess glucose during hyperglycemia? Is insulin the main hormone controlling use of glucose by tissues? What is the role of steroid hormones in the control of glycaemia? and so on.

Perhaps the plausible results of studies trying to answer these questions do not sound “flashy” enough for promotion, or perhaps they are not sufficiently “safe” to merit publication in “prestige” journals, but we need to carry on “risky” studies to get the answers we need in order to understand real problems such as how the substrates provided by the diet are distributed and used. We also need them to understand and fight the negative effects of disorders that right now affect large portions of Humankind. Right now, it is critical to obtain much more basic (and critical) knowledge to understand the nature (and causes) of “inflammation” (obviously, not that defined by Celsus!) that is often used to justify everything that is wrong in MS and related disorders.

## Figures and Tables

**Figure 1 ijms-21-07729-f001:**
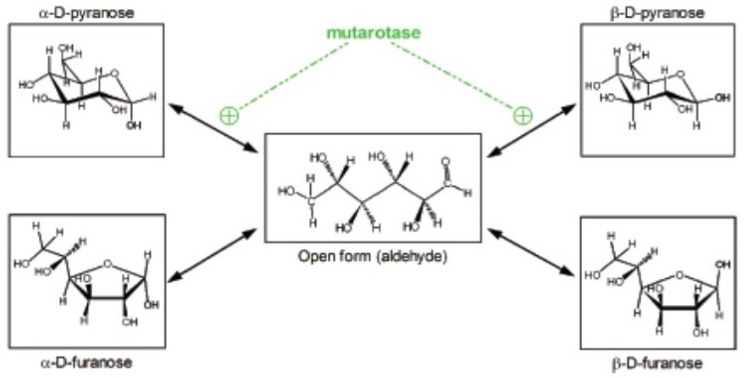
Different structures of the glucose molecule present in biological systems. From Oliva et al. 2019 [10].

**Figure 2 ijms-21-07729-f002:**
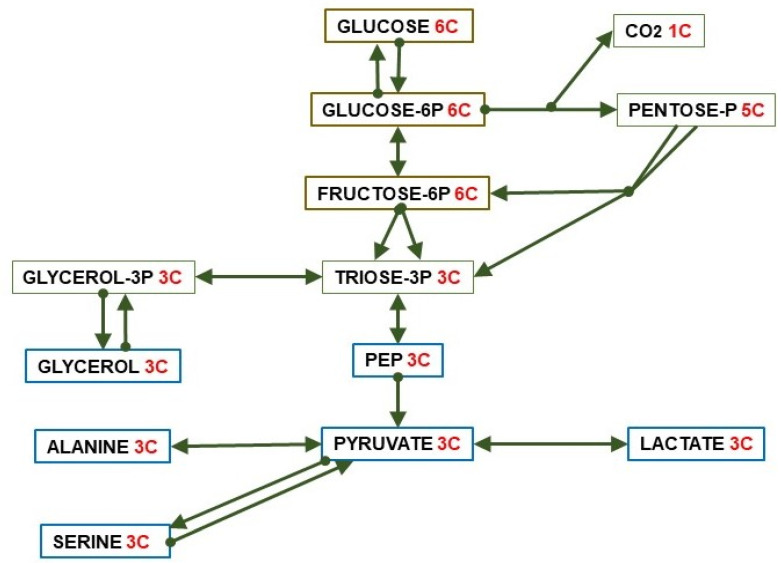
Paths of conversion of glucose (6C) to 3C fragments.

**Figure 3 ijms-21-07729-f003:**
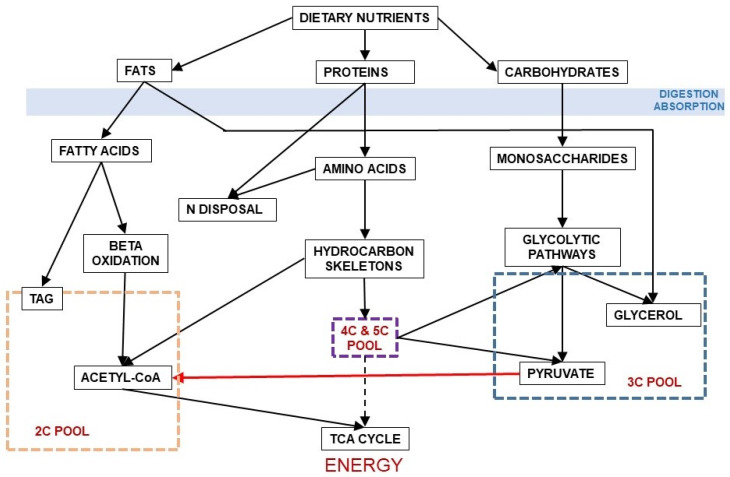
Relationship between the main groups of dietary nutrients driving to the formation of 2C and 3C fragments. In red, the irreversible decarboxylative oxidation path of pyruvate (3C) to acetyl-CoA (2C).

**Figure 4 ijms-21-07729-f004:**
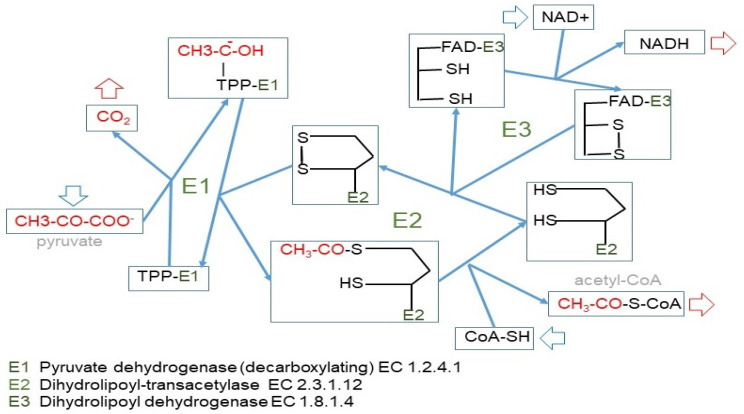
The pyruvate dehydrogenase complex. Upper panel: mechanism of action of the pyruvate dehydrogenase complex, showing why the reaction catalyzed by the E1 enzyme unit is irreversible. Lower panel: kinase/phosphatase regulation of the E1 subunit. Main factors regulating its activation/inhibition.

**Figure 5 ijms-21-07729-f005:**
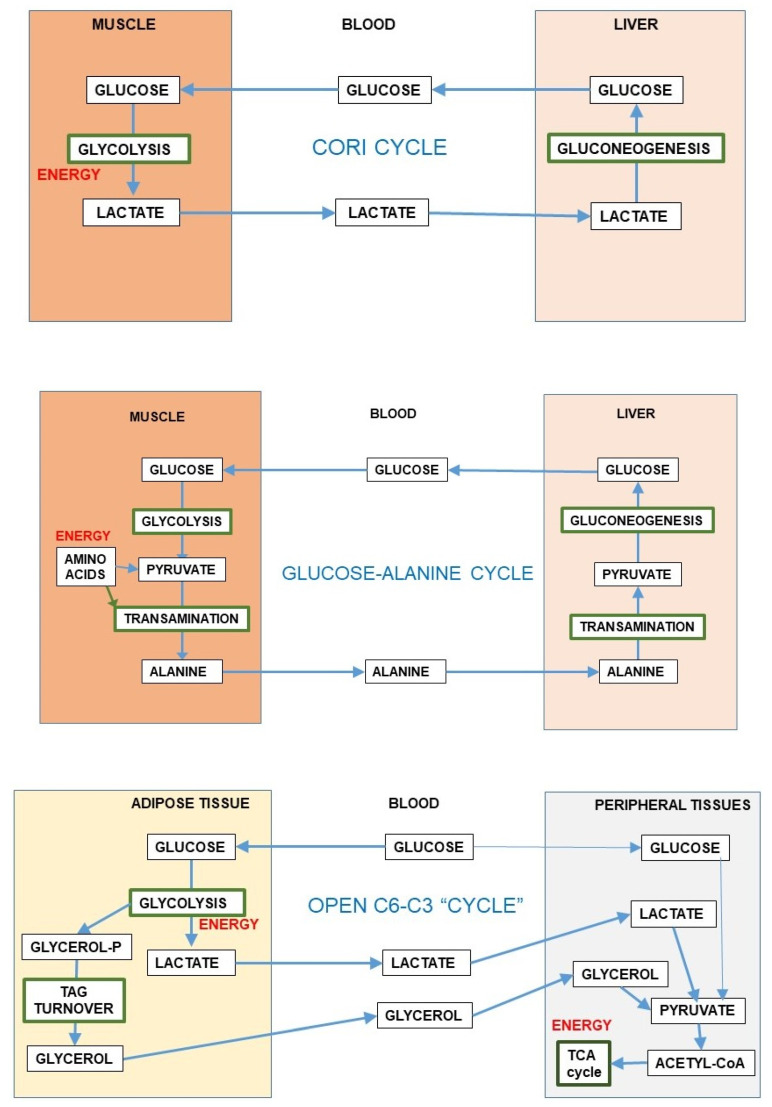
Inter-organ substrate cycles (6C-3C). The first and second panels present the Cori cycle and glucose-alanine cycle [89]. The lower panel shows an “open” inter-organ relationship (6C-3C) such as that found between the adipose tissue and peripheral organs [49].

**Figure 6 ijms-21-07729-f006:**
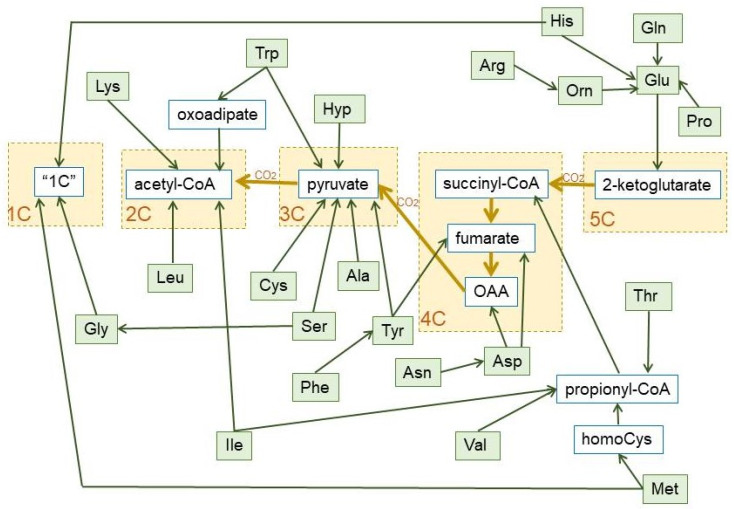
Summary of the final conversion of dietary protein amino acids in 1C, 2C, 3C, 4C, and 5C fragments, which revert essentially into 3C and 2C, during the human catabolism of amino acid hydrocarbon-skeletons. The pathways used to prepare this graph are the most common, including the main alternate pathways [105]. The green lines show the carbon paths for each amino acid, the brown lines represent the relationship between the core of the TCA cycle and pyruvate dehydrogenase (with the loss of one CO_2_ between each substrate group box (marked in yellow). Non-standard abbreviations: Hyp = L-4-hydroxyproline; Orn = ornithine; “1C” = One-carbon fragment donor systems.

**Figure 7 ijms-21-07729-f007:**
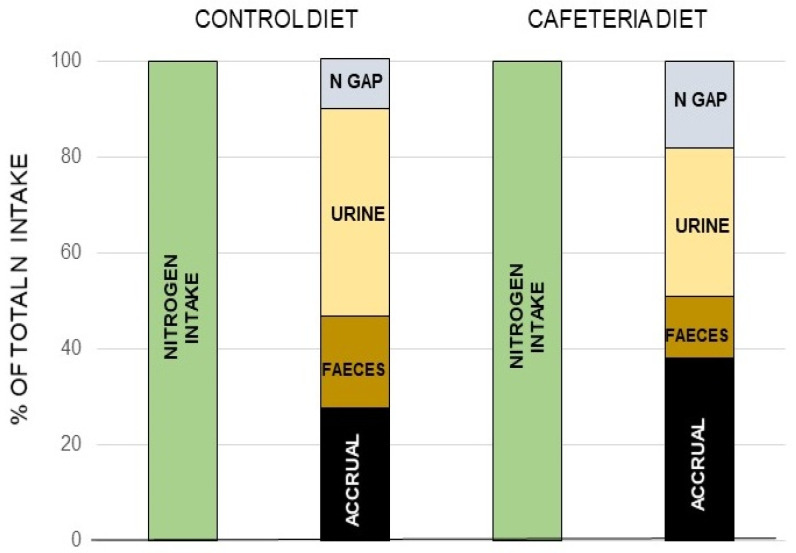
The nitrogen “gap: in rats. A “nitrogen gap” was found when analyzing all the components of the nitrogen balance in young rats: i.e., the N ingested, that excreted by urine and feces as well as the total N accrued, thus proving the existence of a sizeable part of the N excreted not as urea, or through the other possible ways and means shown in Box 1. Different complete (measuring both sides of the N balance equation) studies repeated these findings, dependent on diet, and with a magnitude (in rodents at least) in the range of 10–30% of all nitrogen excreted. Redrawn with data from Esteve et al. [130].

**Figure 8 ijms-21-07729-f008:**
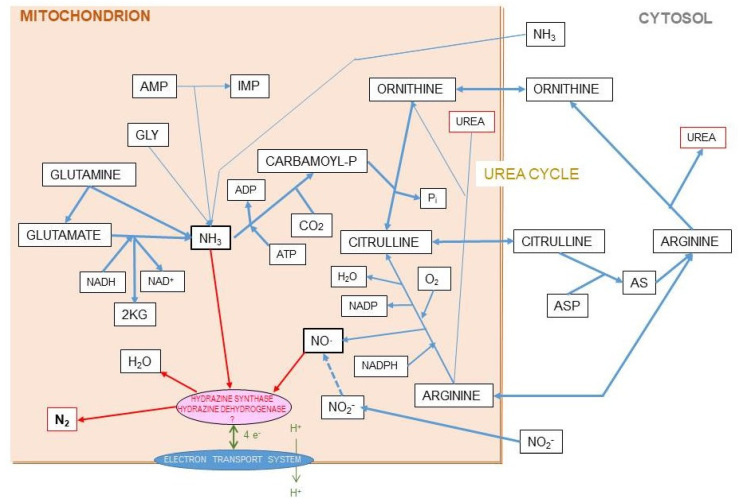
Hypothesis for the origin of ammonia and nitric-oxide generation of nitrogen gas in mitochondria. The coexistence in the liver mitochondrial matrix of both ammonia/ammonium and nitric oxide, within well-regulated pathways, in significant amounts, and related to the urea cycle, suggests the possibility of a reaction which may generate nitrogen gas at the expense of both. This is a speculative hypothesis, which has not been proven so far. In any case it is difficult to justify the presence of both reactants in a relatively high concentration without interacting, since the uncontrolled chemical reaction between them is spontaneous and exergonic. In this hypothesis, we include the findings of Kartal et al. [137] in their study of the mechanism of ammonia oxidation in anammox bacteria, which has an intermediate metabolite between them, hydrazine (N_2_H_4_). The two enzyme activities of the complex hydrazine synthase and hydrazine dehydrogenase are intimately related to membrane of the anammoxosome particle.

**Figure 9 ijms-21-07729-f009:**
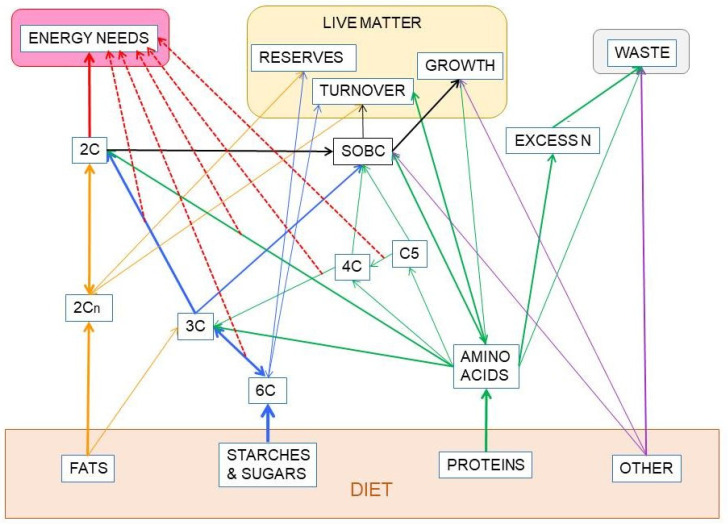
Partition-related substitution of nutrients in the diet to fuel the metabolic processes. This graph intends to explain the absence of a total/complete substitution capability between the three main groups of nutrients: Protein, carbohydrate and fat, which constitute most of our diet, at least from an energy point of view. Solid lines show the direct relationship between the groups of substrates. The large red line emanating from 2C represents the oxidation of acetyl-CoA in the TCA cycle to obtain most of the energy we use. The thinner red dash-lines indicate other main sources of cell ATP, adding up to the sum of energy available to cover our needs. Part of our food is made up of other components (minerals, organic micro-components, fiber, etc.). Their relationships with other compounds are marked with purple lines. SOBC stands for: “synthesis of other body components” from the building blocks provided by the four groups of nutrients analyzed. Their mixed paths have been marked in black. The lines in blue represent the metabolism of carbohydrates, including the part shared with amino acids and lipid. Protein-amino acid paths are marked in green. The lines in dark yellow represent the paths exclusive of lipids.

**Figure 10 ijms-21-07729-f010:**
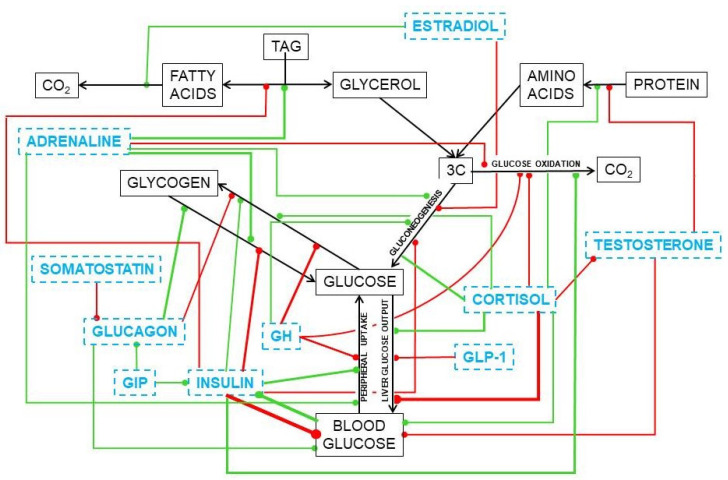
Simplified presentation of the main hormonal mechanisms that control the circulating concentrations of glucose. Black titles and arrows show the basic relationships between glucose and other substrates. Blue titles represent hormones and other regulating agents. Green lines indicate activation or increase effects of the regulator on the signaled path. Red lines show inhibitory or deactivation effects.

**Table 1 ijms-21-07729-t001:** Mass of energy reserves in the human body.

A—Main body reserves mass and its energy equivalents in adult healthy humans.							
**Group**	**Reserve**				**Primary Final Catabolite**		
	**Type**	**Mass, kg**	**Available, kg**	**Energy, MJ**	**MJ 2C**	**MJ 3C**	**2C/3C Ratio**
women	body protein	10.6	2.1	36	4	32	
men		13.5	2.7	45	**5**	41	
women	liver & muscle glycogen	0.38	0.37	6.4	**0**	6	
men		0.48	0.47	8.2	0	8	
women	body fat	12.7	12	445	418	27	
men		16.1	15.3	561	527	34	
women	totals			487	422	65	6.5
men				614	532	83	6.4
B—Whole-body plasma metabolite and energy content equivalents of adult healthy men ^†^.							
**Energy Metabolites**	**mM**	**Whole Plasma mmol**			**Primary Final Catabolite**		
					**mmol 2C**	**mmol 3C**	**2C/3C Ratio**
proteins *	660	170			17	153	
triacylglycerols	1	2.6			25	1	
glucose	5	13			0	26	
amino acids	4.5	11.7			1	11	
glycerol	0.05	0.1			0	0.1	
lactate	0.15	0.4			0	0.4	
NEFA	0.4	1			9	0	
totals					52	192	0.27

^†^ No data have been presented for women, because the scant gender differences in the levels of metabolites, and any correction for plasma volume would not give different values for the final 2C/3C ratio. * Molarity of proteins is expressed as mmol of amino acid residues (mean MW of 105 in the present calculations) [160]. The data in the A part of the Table were taken from Frayn [161], modified with results of [162] and including unpublished data in the calculations. We used standard body weights for men (70 kg) and women (55 kg) to homogenize the data. We assumed that from the whole body protein, only about 20% may be used (allowing recovery) under starvation to cover the body’s energy needs. This datum was calculated from a study of long-term food deprivation in rats [163]. Similarly, an arbitrary limit of 5% minimum of body fat (i.e., not available for energy mobilization) was introduced in the calculations for lack of direct references. No information has been added with respect to more “immediate: energy reserves, such as phosphagens (e.g., creatine-P) and ATP because of their low global energy entity in comparison with the three main storage pools described here. The data in the part B of the Table were calculated only for plasma (i.e., estimated from blood volume minus blood cells, using normal standard concentrations, and a hematocrit value of 43. Here, instead of referring the substrates to their energy content, we used molar concentrations, since they were the primary data.

**Table 2 ijms-21-07729-t002:** Estimation of the 2C/3C relationships of a human standard diet.

Macronutrient	Food Intake, g	Energy Intake		Primary Final Catabolite (mmol/d)				2C/3C Ratio
		%	kJ/d	1C	2C	3C	4C + 5C	
carbohydrate	345	55	5775			3830		
lipid	84	30	3150		5400	100		
protein	94	15	1575	59	413	93	406	
totals		100	10,500	59	5813	4023	406	1.44
					5813	4429		1.31

We used standard consensus-recommended diet compositions [168,169] adapted for a 70 kg healthy adult man, assuming his body weight remains without changes (goals expressed by the WHO and EFSA in the references cited above), that is with no accrual neither losses. The intake was adjusted to 10.5 MJ/d (2500 kcal/d or 122 W). Obviously, the distribution in final metabolites for protein may be subjected to wider changes than lipids and carbohydrate, depending on the protein sources of the diet. We have not found published data for population-wide analysis of amino acids for diet protein composition related to complete and time-sustained normal human diets. In order to get an approximation to the sought data, we used instead data for rat cafeteria diet of previous studies from our research group [170,171], on the assumption that the “cafeteria” diet was devised to mimic the usual consumption of food of young humans late in the past century in urban Westernized settings [154]. In the rat cafeteria diet used, the amino acid residues had a mean MW of 126.

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
