# Peer review of "Dietary Energy Partition: The Central Role of Glucose"

_ijms, 2020, doi:10.3390/ijms21207729_

Round 1

Reviewer 1 Report

This is a comprehensive and insightful review that focused on dietary nutrient dynamics. However, the language of this manuscript is less readable. The authors should consider concisely editing the review further:

  1. Avoiding very long sentences;
  2. Many adverbs were used in the middle of a sentence;
  3. Too many commas were used in a sentence;
  4. Unnecessary single quotes, semicolons and hyphens;
  5. Noticing the use of some words, such as, forfeit and take up;
  6. Avoiding colloquialism;
  7. Revising the title, headings and subheadings, making them easier for the reader to understand;
  8. Defining all abbreviations, including those in figures;
  9. Noticing the use of arrows in figures, such as one-way arrows and double-sided arrows in Figure 2.
  10. More details are needed in figure legends.

Author Response

Comments and Suggestions for Authors:

This is a comprehensive and insightful review that focused on dietary nutrient dynamics. However, the language of this manuscript is less readable. The authors should consider concisely editing the review further:

Thanks for your comments on the focus, extension and insight of our review. The absence of specific criticisms on specific parts of the extensive review, and the global comments drive us to conclude that there are no objections to the review content by the Reviewer. At least no points which we can discuss or modify were pointed at.

However, the Reviewer clearly established that the quality of the English requires extensive editing. This point is complemented with a list of specific points which we are asked to check and correct. It is surprising, however, that Reviewer #2 considered the writing excellent and only asked for correction of typographic errors and misplacing of an abbreviation definition.

In any case, we take at full value the dedicated work of Reviewer #1 and thus we modified the text in accordance with the suggestions provided.

We did not change the core contents of the Review, except for the addition of a caveat on the definition of 3C substrates that was missing in the original version [lines 106-111], in the sense of clarifying that not all three-carbon metabolites were to be considered 3C, as is the case of propionate and D-lactate. This implied the incorporation of another cite (and a second one that was incorrectly cited in the text and thus did not appear in the Bibliography.

The text added (in our opinion) only clarifies, and does not alter the focus, content, meaning and structure of the rest of the manuscript.

With respect to the Edition of the text, we revised the whole review, correcting typographic errors, misleading sentences, shortening them when feasible and actively trying to improve its readability as kindly suggested by the Reviewer. The actions taken with respect to the concrete points listed are additionally described under them. We even found than one figure has a repeated panel. We apologize for the errors, which have been corrected.

The changes introduced in the English language and style (including typographical errors and format defects not present in our initial submission) have been marked using the Microsoft Word revision system (please, see the attached manuscript with corrections). Individual changes, thus, have not been marked with the (new) line numbers when not called on specifically.

  1. Avoiding very long sentences

We did many changes on long sentences, but not on all, by changing semicolons to full stops, limiting parentheses and subordinate sentences (converted into short isolated sentences). However, the nature of the questions analyzed, the enormous disproportion of the literature available for each of the issues discussed and the core of the problem: the incidence of too many factors in the analysis impeding to draw clear-cut conclusions, forced us to maintain some long sentences (such as this one). To be scientifically precise we could not describe complex interactions without including caveats or additional explanations. These changes require either subordinate sentences, punctual explanations, footnotes of the extensive use of adverbs. Nevertheless, we share the Reviewer’s idea of trying to make the text more readable (i.e. more straightforward and less dense), and we did what we could, nor forsaking in the process the necessary precision in the expressions to explain imprecise (by their nature) points.

  1. Many adverbs were used in the middle of a sentence;

This point has been partly explained in point 1. We use adverbs to modulate the absoluteness of the verbs used commonly when describing metabolic situations, this is indeed their reason to exist. This way, the meaning of the text is made clearer, shortening the incertitude that language adds to that derived from the facts being described. In any case, we tried to suppress a few redundant adverbs and to place them in the due order within the sentences.

  1. Too many commas were used in a sentence;

This is a consequence of the long-sentences problem described in point 1. We eliminated the pairs of commas we could safely dispose of, as well as those that were redundant when the sentences were shortened or cut.

  1. Unnecessary single quotes, semicolons and hyphens;

This is an useful literary instrument which calls the attention of the reader on a word or short sentence, we used them here, mainly to indicate the inappropriateness or inadequacy of their use (e.g. ‘inflammation’ for cell pathologic modifications) in the context it is normally used, or to make clear that the specific word meaning in cases of  polysemy, is precisely the one we want to maintain. In any case, we removed some of them because they could be considered unnecessary following the suggestions of the Reviewer. In our opinion, there was not an excess of semicolons. Despite their unjustified scant use in many papers, some of them have been eliminated because of the extensive shortening of long sentences. We did not found any incorrect use of hyphens (except as typographic errors.

  1. Noticing the use of some words, such as, forfeit and take up;

The word ‘forfeit’ has been substituted by ‘ignore’ on L23, Left unchanged on L44 (we found no adequate synonym for the word in this context). On L334 we changed it for ‘eluding’, and on L885 we used ‘forsaking’ instead. Take up is a perfectly normal term in US English (in which this manuscript has been written), its use in reference to incorporating materials into the cell substitutes in its generality, as a global term, means transport, take up, pinocytosis etc. because in the context of the sentence it is unnecessary to describe further (complicate) its structure. No other ‘words’ (please understand our use of single quotes to cite your phrase) were listed and we have no clear idea of their reference, since forfeit is usually in a fairly different set of words (or language style) than taken up.

  1. Avoiding colloquialism;

We are puzzled by this question. Perhaps we can be accused of dense language, the use of some precise albeit not commonly used words, but colloquial writing? Perhaps our use of ‘dark cloud’ on L143 (substituted by ‘potential inconvenient’) may qualify.

  1. Revising the title, headings and subheadings, making them easier for the reader to understand;

We did not see any problem in the title, it just describes the text in two simple short sentences. Our opinion is not to change it, we found that none of the alternatives we discussed were better. As for the headings (there are no subheadings) of the 13 parts of the review, we believe they are just descriptive, albeit short. Perhaps the Reviewer refers to L206, Heading number 5, in which we referred to “glyoxylate dire straits”, now changed to a plain “lack of a glyoxylate shunt”. It was not a colloquialism, since instead of ‘dire straits’ or ‘between Scylla and Charybdis” we would have written ‘between a rock and a hard place’. The sense of a catastrophic metabolic defect with no easy solution we intended seems to be too much literary; we agree with the Reviewer. Scientific language is not –yet— literature stricto sensu, but must clearly convey concepts and ideas. We erased the term and tried to explain better the no-win situation.

  1. Defining all abbreviations, including those in figures;

The only not officially accepted (IUPAC-IUBMB, for instance) abbreviations were in the list already provided except MS (metabolic syndrome) as pointed by Reviewer #2. In the Figures, all abbreviations were standard. In Figure 6, however, the commonly used symbols for hydroxyproline (4-hydroxyproline) Hyp and ornithine Orn, were defined because they are not (to our knowledge) officially recognized as valid amino acid symbols/ abbreviations [L 347-348]. Other not common abbreviations were described in the Figure legends in the first version.

  1. Noticing the use of arrows in figures, such as one-way arrows and double-sided arrows in Figure 2.

The points of arrows in Figure 2 follow the usual system of Biochemistry textbooks: a) one arrowhead line represents a non-reversible path, b) two-arrowhead lines represent reversible paths under physiological conditions, c) two parallel one-arrowhead lines (with opposite directions) represent two different processes allowing the interconversion but via separate paths. We thought that this explanation was unnecessary given the nature of the review and the context of the explanations given in the text.

  1. More details are needed in figure legends.

This manuscript format was changed (as proven by modifications included in it that do not strictly correspond to a ‘peer review’) by the Editorial Office as previous work for its eventual publication. Some long legends of Figures were partially left in the point-10 font of the main text instead of point-9 font of the rest of Figure legends; thus confounding most of these legends with ordinary text. In any case, the Figures are explained within the text, and the sole information needed to include in the legends is, precisely that of colors, abbreviations, general distribution of boxes, units etc. We modified slightly some contents of figure legends but only in the general sense described above of improving readability.

Thanks are given to Reviewer #1 for the time and knowledge devoted to review this manuscript, the keen observations, and dedication invested in helping us to improve our paper if finally accepted.

Reviewer 2 Report

The review "Dietary energy partition. The central role of glucose" is a very in-depth analysis of the current knowledge about dietary energy partition, focusing mostly on the role glucose.

This is a well structured and dense review, very relevant to the field and the writing is excellent.

The final comment is provocative and very pertinent.

Minor point: Metabolic syndrome acronym (MS) appears very late in the text, it is used a few times and afterwards the authors stop using it. This should be corrected. 

Author Response

Thanks for your comments, and specially for your endorsement of our final desperate call for a restructuration of what we do have to investigate.

We are aware that the text is dense, and that there are enormous gaps of knowledge that make this description an exercise (futile?) of equilibrium between unknowns and the current interpretations of what we do with the nutrients we eat.

Since no passages have been marked for revision we interpret that there are no questions with respect to the content of the manuscript.

However, since we have explained in detail in our comments to Reviewer #1, the questions of language, revision of the text and (intended) eradication of typographic errors and incorrect terms has been carried out, and the resulting text is that presented for its eventual final approval and publication.

Please, read our comments with respect to style, grammar and usage of terms already included in our response to the other Reviewer.

Minor point: Metabolic syndrome acronym (MS) appears very late in the text, it is used a few times and afterwards the authors stop using it. This should be corrected.

Thanks for detecting the error. It has been corrected and MS has been used from the start. A similar situation occurred with oxaloacetic acid (OAA), also corrected.

Thanks are given to Reviewer #2 for the time and knowledge devoted to read and review this manuscript, the written general comments, and the dedication invested in helping us to improve our paper if finally accepted.